# Improvement of Tribological Performance of TiAlNbN Hard Coatings by Adding AlCrN

**DOI:** 10.3390/ma15217750

**Published:** 2022-11-03

**Authors:** Yin-Yu Chang, Kai-Chun Huang

**Affiliations:** 1Department of Mechanical and Computer-Aided Engineering, National Formosa University, Yunlin 63201, Taiwan; 2High Entropy Materials Center, National Tsing Hua University, Hsinchu 30013, Taiwan

**Keywords:** multilayer, hard coating, tribological, cathodic arc evaporation

## Abstract

In tribological applications, the degradation of alloy nitride coatings is an issue of increasing concern. The drawbacks of monolayer hard coatings can be overcome using a multilayer coating system. In this study, single-layer TiAlNbN and multilayer TiAlNbN/AlCrN coatings with AlCrN layer addition into TiAlNbN were prepared by cathodic arc evaporation (CAE). The multilayer TiAlNbN/AlCrN showed B1 NaCl structure, and the columnar structure continued from the bottom interlayer of CrN to the top multilayers without interruption. After AlCrN addition, the TiAlNbN/AlCrN coating consisted of TiAlNbN and AlCrN multilayers with a periodic thickness of 13.2 nm. The layer thicknesses of the TiAlNbN and AlCrN were 7 nm and 6.2 nm, respectively. The template growth of the TiAlNbN and AlCrN sublayers stabilized the cubic phases. The introduction of bottom CrN and the TiAlNbN/CrN transition layers possessed com-position-gradient that improved the adhesion strength of the coatings. The hardness of the deposited TiAlNbN was 30.2 ± 1.3 GPa. The TiAlNbN/AlCrN had higher hardness of 31.7 ± 3.5 GPa and improved tribological performance (wear rate = 8.2 ± 0.6 × 10^−7^ mm^3^/Nm) than those of TiAlNbN, which were because the multilayer architecture with AlCrN addition effectively resisted abrasion wear.

## 1. Introduction

TiAlN-based coatings are well known as protective hard coatings and are widely applied in tribological applications [1,2]. By adding Al into TiN, the ternary TiAlN coatings, which are usually deposited using magnetron sputtering and cathodic arc evaporation (CAE), can improve their thermal behavior and, consequently, their in-service tool lives in a tribological environment [2,3]. Particularly, using CAE, a stable NaCl type TiAlN phase is obtained under a high nitrogen concentration, and aluminum atoms replace the titanium atoms to form solid solution TiAlN hard coatings. Multicomponent hard coatings with nanostructures for various tribological applications are expected to have superior mechanical properties under severe corrosive and high temperature environment. The desired mechanical properties can be achieved in hard coating materials based on nitrides of transition metals by complex alloying with elements such as Al, Cr, Nb, Ti and Zr [4,5,6]. Recently, in addition to TiAlN, ternary metal nitride coatings such as TiNbN and TiZrN have been synthesized and possess good mechanical properties than the binary ones [7,8]. By adding Nb into TiAlN, TiAlNbN coatings also have good mechanical and tribological properties and can be used as implant components due to their biocompatibilities [9,10,11]. M. Mikula et al. [10] showed that adding Nb into the TiAlN coatings promoted texture evolution toward the (200) phase, and alloying TiAlN with NbN decreased elastic stiffness while the hardness maintained similar. Nb substitutions are found to improve the ductility of TiAlN without sacrificing hardness and enhance toughness. In addition, AlCrN coatings have attracted great interest from researchers and tool manufacturers due to their good toughness, high temperature oxidation resistance and thermal stability. The dissolution of Al in CrN crystals leads to lattice distortion and grain refinement that contributes to excellent mechanical properties [12,13]. 

To further improve the performance of TiN and TiAlN-based coatings, recently, the drawbacks of monolayer hard coatings can be overcome using a multilayer system [14,15,16]. Nanoscale multilayered coatings have been attracted and studied extensively because of their promising mechanical and tribological properties [17]. Multilayer coatings that consist of different nitride layers and composition-gradient interlayers show improved mechanical properties, such as excellent hardness and adhesion strength due to their specific interfaces [18]. The main factors responsible for improving mechanical and tribological properties include high hardness to elastic modulus ratio, through-thickness composition-gradient properties, resistance to through-thickness cracking by crack deflection between multilayers and adaptive mechanisms from tribo-film formation [19,20]. W.Y.H. Liew [21] showed that the multilayer AlCrN/TiAlN coated tool had higher delamination, abrasion and fracture resistance than the TiAlN single-layer coated tools. Recently, Y. Luo et al. [22] showed that the multilayer coating of AlCrN/TiSiN with nanoscale modulation thicknesses possessed high plasticity index and impact wear resistance to abrasive quartz particles. Presently, there are only a few reports on multilayer coatings composed of TiAlNbN and AlCrN, and particularly their tribological performance has not been widely studied [10,16]. In this study, a high ionization cathodic arc evaporation (CAE) equipped with TiAlNb and AlCr targets was used to deposit the multilayered TiAlNbN/AlCrN with AlCrN layer addition into TiAlNbN. The primary objective of this study was to investigate the mechanical properties and tribological performance of TiAlNbN/AlCrN coatings to evaluate the effect of the AlCrN addition.

## 2. Materials and Methods

### 2.1. Design of Coating Architecture

Two types of TiAlNbN and multilayered TiAlNbN/AlCrN coatings were deposited on Si (100) and cemented carbide (WC- 8 wt.% Co) samples using a CAE system. As shown in Figure 1, targets prepared by TiAlNb and AlCr in atomic proportions 64:26:10 and 70:30, respectively, and pure Cr were used as arc evaporation sources. The composition purity was ≥99.9% for all the targets. The experimental parameters of the deposition process are shown in Table 1. Before the deposition, the chamber was heated to 300 °C and evacuated to base pressure below 1 × 10^−3^ Pa, and all substrates were cleaned by glow discharge under Ar atmosphere at a pressure of 1.6 Pa and a bias voltage of −800 V. The Ar ion etching process was carried out to remove any contamination on the surface of the targets. During the deposition process, a Cr target was used for the metal ion etching, and the film deposition was performed in N_2_ atmosphere. The coating was designed to have a CrN bottom layer, transition TiAlNbN/AlCrN multilayers as composition gradient interlayers, and TiAlNbN and multilayered TiAlNbN/AlCrN were deposited as the top layers. The experimental details are:

A.A CrN was deposited as the bottom layer to enhance the coating adhesion strength to the substrates. The thickness of the CrN was ~200 nm, which was ~1/10 of the total coating thickness. B.Following the bottom layer, transition TiAlNbN/AlCrN multilayers acted as composition gradient interlayers were deposited. The thickness was controlled to be ~200 nm.C.TiAlNbN and multilayered TiAlNbN/AlCrN were deposited as the top layers. For the deposition of multilayered TiAlNbN/AlCrN, the samples were mounted on a rotational holder to undergo a two-fold rotation at a rotation speed of 4 rpm. All coatings were synthesized in N_2_ at a total pressure of 2.7 Pa and a bias voltage of −80 V. The total thickness was ~2 µm by controlling the cathode current of 80 A and deposition time of 50 min.

### 2.2. Coating Characterization

The chemical composition of the coatings was analyzed by electron probe microanalyses (EPMA, JEOL JXA-8530F Plus, JEOL Ltd., Tokyo, Japan). The microstructure of the coatings was identified using X-ray diffraction (XRD, D8 Discover, Bruker Inc., Kanagawa, Japan) with a Cu-Kα X-ray source operated at a voltage of 40 kV and a current of 40 mA. A grazing angle configuration of 2° was used to obtain the crystallographic phases of the deposited coatings. The specimens were scanned with a step size of 0.02°. A scanning electron microscope (SEM, JEOL JSM-7800F, JEOL Ltd., Tokyo, Japan) was used for the observation of the fracture morphology of the coatings. A high-resolution transmission electron microscope (TEM, JEOL JEM-3010, JEOL Ltd., Tokyo, Japan) was employed for microstructural characterization of the coatings in the cross-section geometry. The TEM analyses included the bright field (BF), dark field (DF), HRTEM imaging and selected area electron diffraction (SAED). The cross-sectional TEM samples were prepared using a focused ion beam system (Helios Nanolab 600i System, FEI Co., Hillsboro, OR, USA).

### 2.3. Mechanical Properties and Tribological Analyses

Rockwell indentation tests were conducted to analyze the adhesion strength of the deposited coatings under static contact. The indentation in the study was used to generate cracks, and thus the crack propagation behavior can be observed to evaluate the adhesion strength of the coated samples. The Rockwell indentation tests were carried out at a normal load of 588 N, with a holding time of 4 s, using a conical diamond with a tip radius of 0.2 mm and an aperture of 120°. The nanoindentation tests were performed using a Hysitron Triboindenter (Hysitron TI980 TriboIndenter, Bruker, Billerica, MA, USA) equipped with a Berkovich diamond tip with the indentation depth of 100 nm. Hardness (H) and Young’s modulus (E) measurement of the deposited coatings were accomplished. Here, 5 nanoindentation experiments at different positions were conducted to obtain statistical significance. The H and E values were calculated using the Oliver–Pharr model [23]. The tribological tests of the coatings deposited on the cemented carbide substrate against cemented carbide balls were performed on a ball-on-disk wear testing machine (CSM Instruments, Anton Paar Switzerland AG., Buchs, Switzerland). No lubricant was used in the wear tests. The tribological tests were carried out at room temperature of 28 ± 1 °C with a relative humidity of 65~70%. Experimental parameters were set: a normal load of 5 N, a sliding speed of 30 cm/min and a sliding distance of 500 m. The specific wear rates of the coatings were calculated. The volume loss is equal to the perimeter of the wear track times the cross-sectional area that was profiled by a confocal laser-scanning microscope. The wear tracks were imaged and observed by SEM that complemented with energy dispersive x-ray spectroscopy (JEOL JSM-7800F, JEOL Ltd., Tokyo, Japan). 

## 3. Results and Discussion

### 3.1. Microstructure Characterization 

Table 2 shows the EPMA results that reveal the chemical composition of the deposited coatings. The elemental composition of the TiAlNbN coating was 35.86 ± 0.35 at.% of Ti, 10.61 ± 0.91 at.% of Al, 5.05 ± 0.04 at.% of Nb, and 48.48 ± 1.18 at.% of N. A little lower N content (<50 at.%) was found for the deposited coatings to be substoichiometric. The Al content in the coating was lower than that in the respective Ti_64_Al_26_Nb_10_ targets. This phenomenon can be explained by the mass difference of these elements. Scattering in the N_2_ and re-sputtering from the growing film are more pronounced for lighter elements, such as Al in this case [24]. When the AlCr target was evaporated for deposition at the same time, the TiAlNbN/AlCrN had 18.3 at.% of Ti, 21.4 at.% of Al, 2.8 at.% of Nb, 8.5 at.% of Cr and 49.0 at.% of N. A higher Al content was observed by depositing TiAlNbN and AlCrN.

In Figure 2 and Figure 3, XRD patterns of the TiAlNbN and multilayered TiAlNbN/AlCrN coatings are presented. Peak positions for face-centered cubic (fcc)—TiAlNbN and fcc-AlCrN are marked by dashed lines in Figure 3. All the TiAlNbN and TiAlNbN/AlCrN exhibited a typical fcc rock-salt-type B1 crystal structure. Hexagonal wurtzite-type AlN phases were not detected in the TiAlNbN/AlCrN although the coating possessed a high Al content [25]. Meanwhile, for the multilayer TiAlNbN/AlCrN, distinct TlAlNbN and AlCrN peaks were evident especially at (220) phase. The average crystallite sizes of the coatings were calculated using the Debye–Scherrer formula based on the full width at the half maximum (FWHM) of the (200) peaks. The average grain sizes of the TiAlNbN and TiAlNbN/AlCrN coatings were 7.5 nm and 5.6 nm, respectively. The smaller crystallite size was obtained in the TiAlNbN/AlCrN multilayer coating that resulted from an increase in the interfaces of the layers. From the XRD measurements, the lattice parameters were determined to be 0.4244 nm and 0.4143 nm for the TiAlNbN and AlCrN, respectively. B. Xiao et al. [26] prepared AlCrN using the same AlCr target with an atomic proportion of 70:30, and the lattice parameter of the deposited AlCrN was 0.4113 nm. In this study, the multilayer TiAlNbN/AlCrN showed larger lattice parameter than that of the AlCrN coating, which was considered to be due to the lattice distortion by introducing a multilayer structure [27]. The template growth of the cubic TiAlNbN and AlCrN layers stabilized the cubic phases. Similar results were also observed for AlTiN/AlCrBN and AlTiN/CrTiSiN multilayers in previous studies [28,29]. Recently, a similar TiAlN/ReN multilayer coating with template growth of cubic crystalline structures was deposited by reactive DC magnetron sputtering, and the multilayer TiAlN/ReN possessed a beneficial effect on corrosion [30].

Figure 4 shows the cross-sectional SEM micrographs of the TiAlNbN and multilayered TiAlNbN/AlCrN coatings. The deposited TiAlNbN consisted of a CrN bottom layer, a TiAlNbN/CrN transition layer, and a top TiAlNbN layer, while the deposited TiAlNbN/AlCrN consisted of a CrN bottom layer, a TiAlNbN/CrN transition layer, and a top TiAlNbN/AlCrN. From the SEI, typical columnar growth was observed for the both coatings. For the TiAlNbN, the columnar grains grew from the bottom CrN through the TiAlNbN/CrN transition layer to the top TiAlNbN. The columns of the top TiAlNbN became more obvious and larger than those of the bottom CrN and the TiAlNbN/CrN transition layer. The TiAlNbN/AlCrN coating exhibited a denser structure than the TiAlNbN coating resulted from the effect of the introduction of AlCrN and multilayer interruption, which inhibited the coalescence of columnar growth. Similar results of dense columnar structures can be observed in the AlCrN/AlTiSiN multilayer coatings with periodic thickness of ~11.6 nm [31]. The dense and multilayered structure had the effect of combining the concept of alloying of AlTiN by adding Si and the application of a multilayer architecture on the oxidation resistance and improved the mechanical properties. Y. Li et al. [32] also found the mechanical and tribological properties can be improved by the design of a dense-and-compact multilayer coating which consisted of a bottom CrN layer, and a middle CrTiAlSiN with columnar structure growth, and an outmost superlattice WCrTiAlN layer. In this study, alternating deposition of TiAlNbN and AlCrN monolithic layers hindered the growth of columnar grains, and reduced internal defects of pin holes and large columnar grain boundaries, resulting in better compactness of TiAlNbN/AlCrN coating. 

Figure 5 shows the high magnification cross-sectional HRTEM micrograph of the top TiAlNbN/AlCrN coating and corresponding fast Fourier transform (FFT) images of each AlCrN and TiAlNbN layers. Similar to the results of XRD, hexagonal wurtzite-type AlN phases were not detected in the TiAlNbN/AlCrN. Although the multilayer coating exhibited a slightly higher percentage of Al than Ti, the coating was far from the possible spinodal decomposition and formation of hexagonal phases of AlN [33]. The TiAlNbN/AlCrN exhibited columnar growth multilayer structure with dark and bright layers alternating in growth direction. Using the different cathodic arc sources (AlCr and TiAlNb), the multilayer structures were formed by alternate deposition of each TiAlNbN and AlCrN. AlCrN. The dark layer was TiAlNbN and the bright layer was AlCrN, respectively. The layer thicknesses of the TiAlNbN and AlCrN were 7 nm and 6.2 nm, respectively. The periodic thickness of the TiAlNbN/AlCrN was 13.2 nm. The periodic thickness in nanoscale is the most important parameters in multilayer hard coatings. The enhancement of mechanical properties generally occurs in a narrow range of 5–15 nm due to dislocation blocking by layer interfaces and strain effects at layer interfaces [34,35]. 

Under the same cathode current (80 A) used, the smaller layer thickness of AlCrN showed that the deposition rate of AlCrN was lower than that of TiAlNbN. The inset HRTEM micrograph showed that the lattice fringes grew cross the layer interfaces. The interplanar spacing values 0.2463 nm and 0.2386 nm measured from FFT revealed that the continuous growth of the same fcc TiAlNbN and AlCrN with (111) phases, respectively. Therefore, the HRTEM micrograph verified the coherent interfaces of TiAlNbN/AlCrN multilayer with single-phase fcc structure. The continuous growth of TiAlNbN/AlCrN multilayers with the same fcc structure was not interrupted by the sublayer interfaces. The TiAlNbN/AlCrN multilayer structure with coherent interfaces could be beneficial to the mechanical and tribological performances [36,37]. 

### 3.2. Mechanical Properties and Tribological Performance

The adhesion strength of the hard coating is an important mechanical property and can directly affect the service life of coating products. Figure 6 shows the optical micrographs of indentations of the deposited TiAlNbN and multilayered TiAlNbN/AlCrN coatings on cemented carbide (WC- 8 wt.% Co) samples. Spalling at the interfaces of both the indentation impress and the coating indicates that the adhesion strength is low. In this study, no coating spallation was observed on the areas out the indentation of both the coated samples. It was found that only radial cracks were observed (as shown by arrows) on both the TiAlNbN and multilayered TiAlNbN/AlCrN coated samples. Careful observations showed that no significant peeling-off phenomenon occurred, which indicated that both the TiAlNbN and multilayered TiAlNbN/AlCrN samples had high adhesion strength, which was important for the tribological applications [38,39]. To improve the adhesion strength between the hard nitride coating and soft metallic substrate, functionally graded materials (FGMs) can be introduced as a transitional layer in coating systems, known as functionally graded coatings (FGCs) [40]. In recent years, FGCs have played a vital role in thin film and coating engineering applications due to their distinctive features through graded characteristics. In this study, the introduction of bottom CrN and the TiAlNbN/CrN transition layers possessed composition-gradient that improved the adhesion strength of the coatings, and reduced damage and cracks in the coating under loading [41]. H.K. Kim et al. [42] proved that the CrZrN coating structure with proper gradient CrN_x_ interlayers induced a smooth transition of stress in the coating under a loading condition. The adhesion strength of the coatings could be improved significantly by structuring the coating with an optimal gradient interlayer. Y.W. Lin et al. [43] and M.W.J. Liu et al. [44] also confirmed the importance of TiN/Ti interlayers in TiZrN coatings. The residual stress in TiZrN coatings could be relieved by the TiN/Ti interlayers.

Table 3 shows the hardness (H), Young’s modulus (E), H/E, H^3^/E^2^, coefficient of friction (COF) and wear rate of the TiAlNbN and multilayered TiAlNbN/AlCrN coatings compared with other hard coatings published in the previous studies. The hardness of the deposited TiAlNbN was 30.2 ± 1.3 GPa. For the typical Ti_1-x_Al_x_N hard coating with different Al contents, the highest hardness was obtained for the Ti_1-x_Al_x_N with x = 0.6, and the decrease in hardness with decreasing Al content [45]. For the Ti_1-x_Al_x_N with x~0.2 similar to the present TiAlNbN in this study, the hardness was lower than 25 GPa. In this study, the deposited TiAlNbN possessed higher hardness due to the addition of a small fraction of Nb (~5 at.%). M. Mikula et al. [10] showed that the addition of Nb in TiAlNbN could retain high hardness while reducing stiffness and improving ductility, indicating toughness enhancement. Similar results were also found in the study of Y.H. Chen et al. [46]. 

A previous study showed that the hardness of the AlCrN was ~27 GPa, which the coating was deposited by the CAE using the same AlCr target in an atomic proportion of 70:30 [47]. Although the softer AlCrN was added, the TiAlNbN/AlCrN exhibited higher hardness value (31.7 GPa) compared to the TiAlNbN and AlCrN. The H/E and H^3^/E^2^ ratios were used to represent resistance against elastic strain to failure and resistance to plastic deformation, respectively [48]. The higher H/E and H^3^/E^2^ ratios were found for the multilayer TiAlNbN/AlCrN. It is clearly observed that the hardness and elastic modulus of the TiAlNbN/AlCrN coating are clearly higher than those of the monolayer coatings. Similar observations are also found in the TiN/CrN, TiN/ZrN and TiN/WN multilayer systems [49]. The observed improvement of the mechanical parameters, such as hardness and following H/E or H^3^/E^2^ ratios in the studied multilayer coatings, might most likely be related to the decrease in bilayer thickness in nanoscale and subsequently lowered crystalline/grain sizes. This led to an increase in interface volume and boundaries as pinning points in material [50]. The existence of the high fraction interface structure can achieve higher hardness to obtain higher H/E and H^3^/E^2^ ratios. A recent study by H. Guo et al. [51] showed that the CrAlN/CrN multilayer coating with the advantages of high hardness and high fracture toughness could improve the erosion resistance. The interface in the multilayer coating effectively hindered the propagation of the crack, and therefore increased the erosion resistance of the coatings. Erosion resistance of the multilayer coatings was positively related to the hardness and fracture toughness. Therefore, the improvement of the mechanical properties, such as hardness, H/E, and H^3^/E^2^ ratios in the multilayer TiAlNbN/AlCrN coating, was related to the interlayer design and multilayered structure in nanoscale [52,53]. G.S. Fox-Rabinovich et al. [54] showed that TiAlCrSiYN/TiAlCrN coatings with modulating chemical composition but similar characteristics of the alternating nanolayers improved mechanical properties of the TiAlCrSiYN-based coatings. The H/E and H^3^/E^2^ ratios were 0.087 and 0.214 GPa, respectively. Although the TiAlCrSiYN/TiAlCrN coatings had more alloying elements in multilayers, the TiAlNbN/AlCrN exhibited higher H/E (0.090) and H^3^/E^2^ (0.254 GPa) ratios. It indicated that the H/E and H^3^/E^2^ ratios were not increased by complex multicomponent systems.

The multilayer structure can effectively dissipate the fracture energy, and reduce the possibility of crack propagation to enhance the resistance to plastic deformation and abrasion wear [37,55]. M-R. Alhafian et al. [2] had shown that the TiAlN coating deposited by CAE exhibited high friction coefficient (0.7~0.8) and wear rate which correlated to the severe adhesive wear. The results of the ball-on-disk measurements performed in this study are shown in Table 3 and Figure 7. For the TiAlNbN coating, the friction coefficient increased from 0.6 to 0.8, and its average coefficient of friction (COF) value was 0.73. The steady-state friction is governed by the shear resistance of the tribolayer and the wear interaction with the counterpart [15,56]. The lower average COF (0.68 ± 0.03) and wear rate (8.2 ± 0.6 × 10^−7^ mm^3^/Nm) were obtained for the TiAlNbN/AlCrN. The wear resistance of the TiAlNbN/AlCrN coating depended on its mechanical properties. A previous study of TiAlN/TiSiN multilayer coatings with high hardness of 33–36 GPa showed the COF was as high as 0.7–0.8, and the wear rate was ~5 × 10^−6^ mm^3^/Nm [15]. In comparison, the TiAlNbN/AlCrN prepared in this study possessed better wear resistance than the TiAlN/TiSiN multilayer coatings. A. Leyland and A. Matthews [57] had proved the significance of the H/E ratio in wear control of hard coatings to optimize tribological performance. S.A. Plotnikov et al. [58] studied the wear behavior of multilayered TiC_x_/Ti/a-C coatings with different H^3^/E^2^ ratios, and they showed that the wear resistance of multilayered TiC_x_/Ti/a-C coatings increased with increasing H^3^/E^2^ ratio. As compared to the wear resistances of TiAlN and AlCrN coatings which were studied by E. Kaya et al. [59], although the hardness values of TiAlN and AlCrN were similar to the present TiAlNbN/AlCrN, the TiAlNbN/AlCrN with higher H/E and H^3^/E^2^ ratios showed better wear resistance.

Due to testing against hard WC-Co balls, abrasion was the predominant wear mechanism. Hard WC asperities penetrated into the coatings that resulted in their scratching and chipping. The wear track SEM images of the TiAlNbN and multilayer TiAlNbN/AlCrN coatings as well as EDS elemental mapping are shown in Figure 8. For the multilayer TiAlNbN/AlCrN coating, the middle band of the wear surface on the wear track was clean and smooth. The EDS mapping at the wear track area of the TiAlNbN showed distinct enrichment of W, which exhibited coating delamination, indicated by the exposure of the WC-Co substrate. The hard wear debris acted as an abrasive increasing the wear rate. In contrast, the TiAlNbN/AlCrN with higher H, H/E and H^3^/E^2^ ratios did not exhibit delamination that confirmed the stronger bonding between the layers resulted in the better wear resistance. Our previous study also showed that multilayer AlTiN/CrN/ZrN coatings with CrN layer addition could possess good wear performance due to the multilayer architecture with CrN addition that effectively released the internal stress to resist abrasion wear [14]. Y.X. Ou et al. [52] also showed that a multilayer Si_3_N_4_/CrN coating had high hardness and toughness using CrN introduction, and possessed cracking resistance to improve wear resistance. In this study, the results showed that the multilayer TiAlNbN/AlCrN coating with AlCrN addition into TiAlNbN possessed good wear resistance due to high resistance to plastic deformation that scaled with H^3^/E^2^ ratio. This reduced the probability of initiated cracks on the wear surface. The dense columnar TiAlNbN/AlCrN multilayers were thought to enhance the mechanical properties and wear performance. Given the wide range of materials and methods used to fabricate hard coatings for tribological applications, by selecting the proper combination of materials and design strategy, multilayer coatings, such as the TiAlNbN/AlCrN in this study, can be adapted to reduce friction and increase wear resistance of mechanical components in a variety of tribological applications [60]. 

**Table 3 materials-15-07750-t003:** Hardness (H), Young’s modulus (E), H/E, H^3^/E^2^, coefficient of friction and wear rate of the TiAlNbN and multilayered TiAlNbN/AlCrN coatings compared with other hard coatings published in the previous studies.

Coatings	Hardness (GPa)	Young’s Modulus (GPa)	H/E	H^3^/E^2^	Coefficient of Friction	Wear Rate × 10^−7^ (mm^3^/Nm)	References
TiAlNbN	30.2 ± 1.3	351 ± 14	0.086	0.224	0.73 ± 0.06	23.1 ± 0.5	This study
TiAlNbN/AlCrN	31.7 ± 3.5	354 ± 21	0.090	0.254	0.68 ± 0.03	8.2 ± 0.6	This study
TiAlN	30.7~31.4	520~528	0.06	0.108	0.7~0.8	-	[2]
TiAlN	31	359	0.086	0.231	0.5	~45	[59]
AlCrN	32	363	0.088	0.249	0.38	~32	[59]
TiAlNbN	28~31	∼442	0.07	0.152	-	-	[10]
TiAlN/TiSiN	33~36	~480	0.075	0.202	0.7~0.8	~50	[15]
TiAlCrSiYN/TiAlCrN	28.4 ± 4.8	327 ± 36	0.087	0.214	-	-	[54]

## 4. Conclusions

The influence of AlCrN addition to TiAlNbN on the structural and mechanical properties of multilayered TiAlNbN/AlCrN coatings was investigated. Improved adhesion strength and wear performance of the multilayered coatings could be achieved using the appropriate combinations of CrN bottom layer and intermediate TiAlNbN/AlCrN layers. The TiAlNbN coatings showed a multi-phase structure dominated by a face-centered cubic (fcc) structure, with an increase in the intensity of a (111) peak. Using the different cathodic arc sources (AlCr and TiAlNb), the multilayer structure of TiAlNbN/AlCrN was formed by alternate deposition of each TiAlNbN and AlCrN. The layer thicknesses of the TiAlNbN and AlCrN were 7 nm and 6.2 nm, respectively. Although the softer AlCrN was added, the existence of the high fraction interface structure of TiAlNbN/AlCrN can achieve higher hardness to obtain higher H (31.7 ± 3.5 GPa), H/E (0.090) and H^3^/E^2^ (0.254) ratio than those of TiAlNbN. The multilayer structure can effectively dissipate the fracture energy, and reduce the crack propagation to enhance the resistance to plastic deformation and abrasion wear. The lower average COF (0.68 ± 0.03) and wear rate (8.2 ± 0.6 × 10^−7^ mm^3^/Nm) were obtained for the TiAlNbN/AlCrN compared with other hard coatings such as TiAlNbN, TiAlN and TiAlN/TiSiN. The design of TiAlNbN/AlCrN multilayer coatings with periodic thickness in nanoscale can maintain high hardness and improve tribological performance. In summary, the mechanical and tribological properties have been improved using the design of TiAlNbN/AlCrN multilayer coatings with composition-modulation geometries. Future work on different composition-modulation geometries and chemical constitutions can be considered when designing wear-resistant nanomultilayer coatings.

## Figures and Tables

**Figure 1 materials-15-07750-f001:**
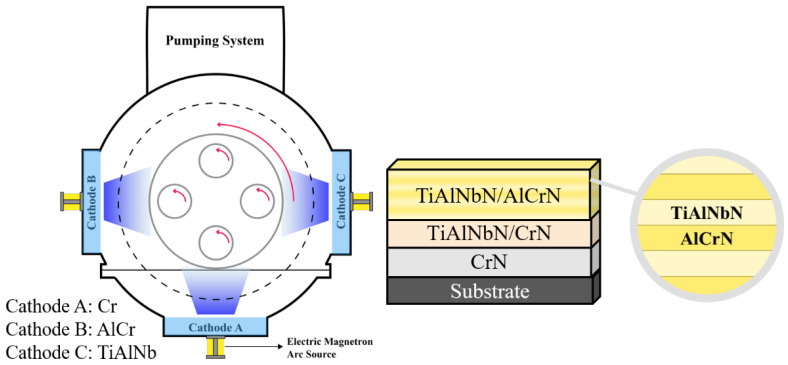
Schematic representation of the deposited TiAlNbN/AlCrN coatings using a cathodic arc system with multi-cathodes.

**Figure 2 materials-15-07750-f002:**
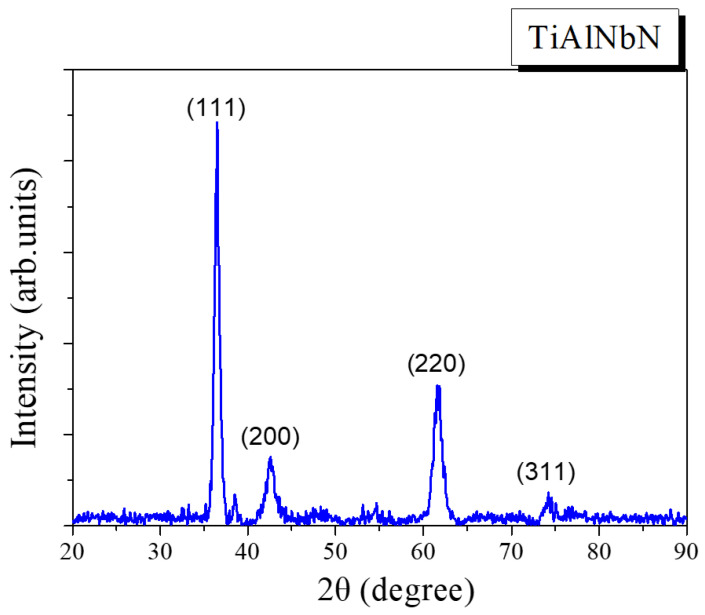
Glancing angle X-ray diffraction pattern of the deposited TiAlNbN coating.

**Figure 3 materials-15-07750-f003:**
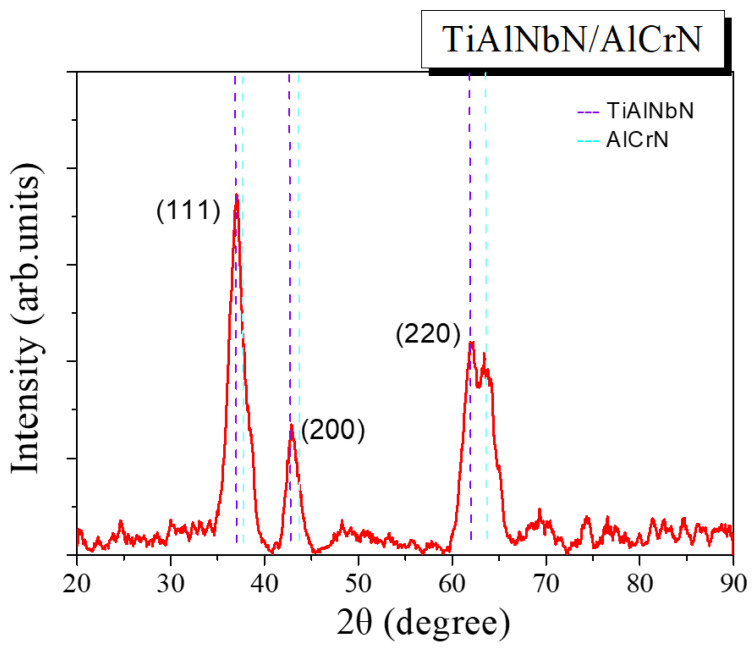
Glancing angle X-ray diffraction pattern of the deposited TiAlNbN/AlCrN coating.

**Figure 4 materials-15-07750-f004:**
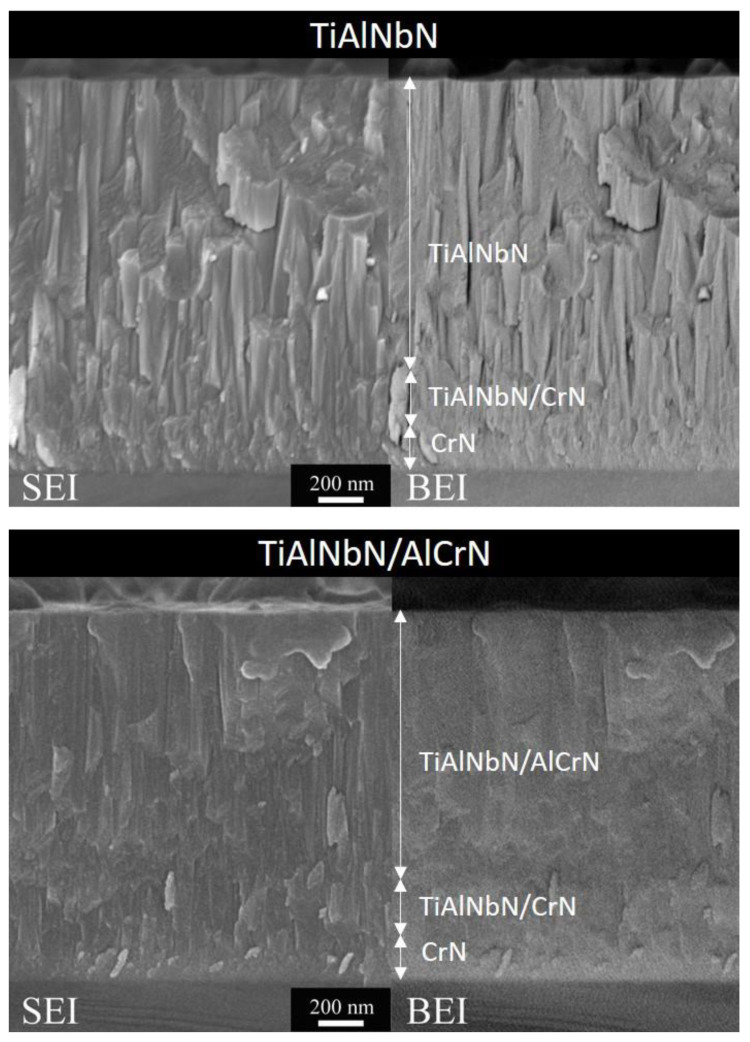
Cross-sectional SEM micrographs of the deposited TiAlNbN and multilayered TiAlNbN/AlCrN coatings showing secondary electron images (SEI) and backscattered electron images (BEI).

**Figure 5 materials-15-07750-f005:**
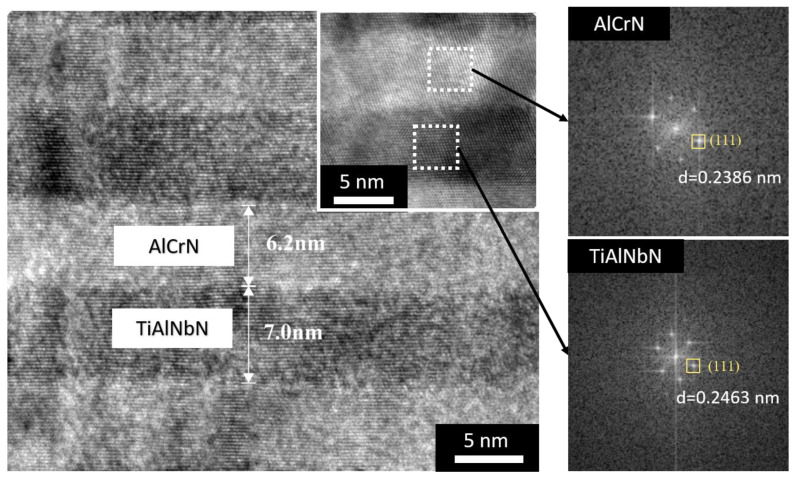
High magnification cross-sectional HRTEM micrograph of the top TiAlNbN/AlCrN coating (**left**) and corresponding FFT images (**right**) of each AlCrN and TiAlNbN layers.

**Figure 6 materials-15-07750-f006:**
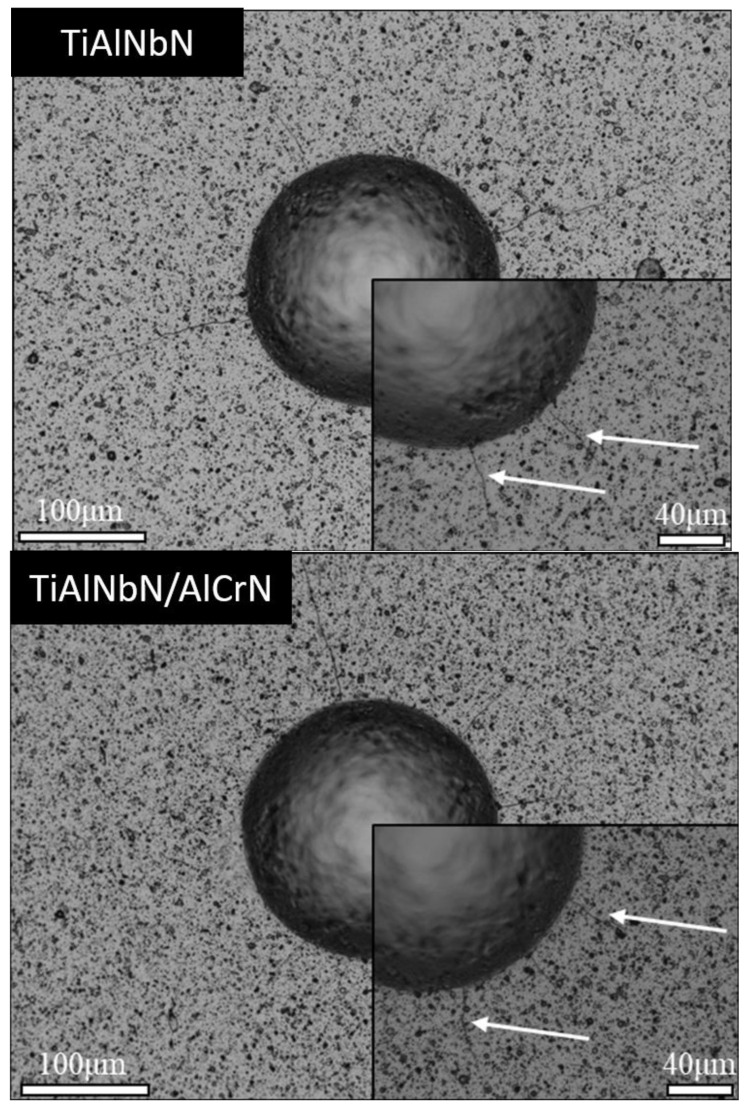
Optical micrographs of Rockwell indentations of the deposited TiAlNbN and multilayered TiAlNbN/AlCrN coatings on cemented carbide (WC- 8 wt.% Co) samples measured using Rockwell indentation. The white arrows indicate the radial cracks after indentation.

**Figure 7 materials-15-07750-f007:**
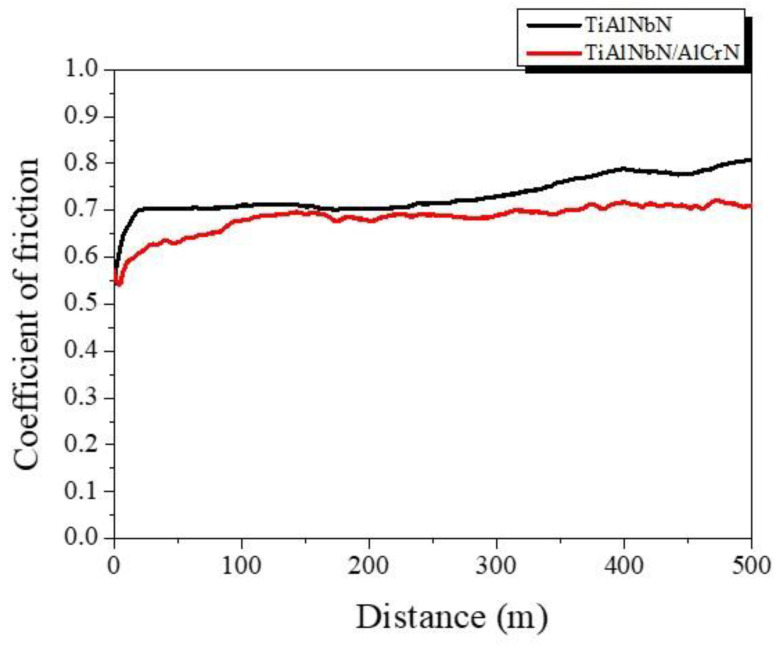
The coefficient of friction of the deposited TiAlNbN and multilayered TiAlNbN/AlCrN coatings on cemented carbide (WC- 8 wt.% Co) samples against cemented carbide balls as a function of sliding distance in the dry environment.

**Figure 8 materials-15-07750-f008:**
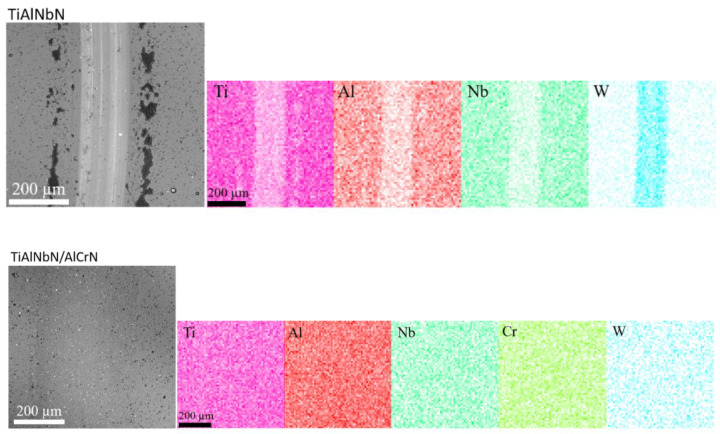
The SEM surface morphologies and corresponding EDS elemental mapping of the wear tracks of the TiAlNbN and multilayered TiAlNbN/AlCrN coatings after sliding distance of 500 m.

**Table 1 materials-15-07750-t001:** Depositing parameters of the TiAlNbN/AlCrN coatings.

Parameters	Values
CAE target	Cr, Ti_64_Al_26_Nb_10_, Al_70_Cr_30_ (100 mm in diameter)
Distance between target and substrate (mm)	180
Base pressure (Pa)	<1.0 × 10^−3^
Bias voltage at ion cleaning stage (V)	−800
Ar pressure at ion cleaning stage (Pa)	1.6
Reactive N_2_ pressure (Pa)	2.7
Deposition time (min) at bottom-layer stage	5
Deposition time (min) at transition-layer stage	5
Deposition time (min) at top layer stage	50
Cathode current (A)	80
Bias voltage during deposition (V)	−80
Substrate temperature (°C)	300 ± 20
Rotational speed of the substrate (rpm)	4

**Table 2 materials-15-07750-t002:** Chemical composition of the deposited TiAlNbN and multilayered TiAlNbN/AlCrN coatings measured by EPMA.

	Al (at.%)	Ti (at.%)	Cr (at.%)	Nb (at.%)	N (at.%)
TiAlNbN	10.61 ± 0.91	35.86 ± 0.35	-	5.05 ± 0.04	48.48 ± 1.18
TiAlNbN/AlCrN	21.39 ± 0.88	18.56 ± 0.58	8.38 ± 0.04	2.69 ± 0.07	48.98 ± 1.51

## Data Availability

Not applicable.

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
