# Peer review of "Improvement of Tribological Performance of TiAlNbN Hard Coatings by Adding AlCrN"

_materials, 2022, doi:10.3390/ma15217750_

Round 1
Reviewer 1 Report
Review comments on materials-1983508
In this work, single-layer TiAlNbN and multilayer TiAlNbN/AlCrN coatings with AlCrN layer addition into TiAlNbN were prepared by cathodic arc evaporation (CAE) and investigated. This manuscript will be considered for acceptable after revised based on the following comments:
1. Abstract must be improve with more data
2. In the current state, there are some typographical errors. Therefore, the authors are advised to recheck the whole manuscript for improving the language and structure carefully.
3. The introduction section is very short and poorly described. It doesn't present the reference to the manuscript scope. In the introduction section, authors should make an in-depth literature review concerning the application of advanced compounds and materials in various fields. Introduction has deficiency citation to valuable works published before such as: ACS Appl Mater Interfaces 2021 Jul 7;13(26):31066-31076; Ultrasonics Sonochemistry 82 (2022) 105892; authors should be cite to these works to improve introduction section.
4. Purity of using materials must be clear
5. And the structure of the manuscript might need a major adjusting for a better understanding.
6. The writing logic of characterization analysis is not clear.
7. Redesign the methods chapter the way so anybody can repeat your procedures, like a recipe
8. Results and discussion: - To increase the scientific value of the manuscript Authors should consider extension of the all results section with comparison of obtained results with the results described in previous publications.
9. The figures in whole manuscript are of poor quality, which leads to difficulty in reading and understanding the manuscript. Please refine
10. What is the main significance of paper in comparison to other published works?
11. This work should be compared with the other work in Table form.
12. Conclusion must be improve with obtaining data and also describe about future
13. The authors should determine from XRD data the average crystallite size of the products by Scherer equation.
14. The authors should prepare all tables with better quality.
Author Response
Dear Editor and Reviewer:
Thank you very much for spending your precious time in processing our manuscript. I deeply appreciate for your kindness to consider acceptance of our work and for the constructive comments on our manuscript. We have carefully revised the manuscript according to the suggestion raised by the Reviewers. The English grammar of this manuscript is carefully revised with help from a person proficient in the English language according to referee’s suggestions. The changes and modifications of the manuscript are highlighted in RED color through the text of the revised manuscript. Additional response regarding the referee’s comments is provided in a WORD file (author-coverletter-23239639.v1.docx).

Reviewer 2 Report
1) Chapter "Discussion" is missing in the article.
2) What do the authors think, why is the coefficient of friction so high? (In fig. 7, Cof=min. 5.5 )
Author Response
Dear Editor and Reviewer:
Thank you very much for spending your precious time in processing our manuscript. I deeply appreciate for your kindness to consider acceptance of our work and for the constructive comments on our manuscript. We have carefully revised the manuscript according to the suggestion raised by the Reviewer. The changes and modifications of the manuscript are highlighted in RED color through the text of the revised manuscript. Additional response regarding the referee’s comments is provided in a WORD file (author-coverletter-23240923.v1.docx).

Reviewer 3 Report
This work is focused on researches about improvement of tribological performance of TiAlNbN hard coatings by adding AlCrN. It shows some interesting results and discussions. It can be considered for publication after addressing the following comments:
1) Please correct "…bias voltage of -800V" as "bias voltage of -800 V".
2) Please correct "a little lower N content (< 50 at.%) was found for the deposited coatings to be substoichiometric" as "A little lower N content (< 50 at.%) was found for the deposited coatings to be substoichiometric".
3) Please correct "Peak positions for face-centered cubic (fcc)-TiAlNbN and fcc-AlCrN are marked by dashed lines in Fig.3" as "Peak positions for face-centered cubic (fcc) - TiAlNbN and fcc-AlCrN are marked by dashed lines in Fig. 3".
4) Please indicate at what temperature and humidity the tribological tests were carried out. Please specify at how many samples were investigated for friction tests.
5) Please state version, manufacturer, city and country from where a ball-on-disc tribometer has been sourced.
Author Response
Dear Editors and Reviewer:
Thank you very much for spending your precious time in processing our manuscript. I deeply appreciate for your kindness to consider acceptance of our work and for the constructive comments on our manuscript. We have carefully revised the manuscript according to the suggestion raised by the Reviewer. In this 2nd revision edition, the English grammar of this manuscript is carefully revised again with help from a person proficient in the English language according to referee’s suggestions. The changes and modifications of the manuscript are highlighted in RED color through the text of the revised manuscript. Additional response regarding the referee’s comments is included as attached WORD file.

Round 2
Reviewer 1 Report
Recommendation: Reject
In this manuscript, "Improvement of tribological performance of TiAlNbN hard coatings by adding AlCrN”, the whole quality is not high. The discusstion is not deep and clear. So it is not recommended to be accepted.
Author Response

(The authors gave the same response as above.)
